# Structural basis for stabilisation of the RAD51 nucleoprotein filament by BRCA2

Robert Appleby [1], Luay Joudeh[1], Katie Cobbett[1] & Luca Pellegrini [1] ✉

The BRCA2 tumour suppressor protein preserves genomic integrity via interactions with the DNA-strand exchange RAD51 protein in homology-directed repair. The RAD51-binding TR2 motif at the BRCA2 C-terminus is essential for protection and restart of stalled replication forks. Biochemical evidence shows that TR2 recognises filamentous RAD51, but existing models of TR2 binding to RAD51 lack a structural basis. Here we used cryo-electron microscopy and structure-guided mutagenesis to elucidate the mechanism of TR2 binding to nucleoprotein filaments of human RAD51. We find that TR2 binds across the protomer interface in the filament, acting as a brace for adjacent RAD51 molecules. TR2 targets an acidic-patch motif on human RAD51 that serves as a recruitment hub in fission yeast Rad51 for recombination mediators Rad52 and Rad55-Rad57. Our findings provide a structural rationale for RAD51 filament stabilisation by BRCA2 and reveal a common recruitment mechanism of recombination mediators to the RAD51 filament.

The *BRCA2* tumour suppressor gene has a central role in maintenance of genomic stability[1,2]. Familial or sporadic mutations that impair BRCA2 function predispose to an increased lifetime risk of breast, ovarian and other cancers[3]. *BRCA2* knockouts are embryonically lethal[4], but experiments with hypomorphic *BRCA2* alleles revealed that BRCA2 functions in homology-directed repair (HDR) of DNA double-strand breaks (DSBs)[5–7]. Subsequent research uncovered a role for BRCA2 and RAD51, as well as for Fanconi Anemia proteins[8] and RAD51 paralogues[9], in protecting DNA replication forks arrested by spontaneous or induced lesions from nucleolytic degradation[10–17]. The sensitivity of BRCA2-deficient tumours to inhibitors of the DNA repair enzyme PARP has been exploited therapeutically[18], emphasising the importance of elucidating the mechanisms of such synthetic lethality and the ensuing resistance following treatment.

Both functions of BRCA2 in DSB repair and replication require specific interactions with the DNA-strand exchange RAD51 protein. In HDR, BRCA2 loads RAD51 onto the single-stranded overhangs of resected DNA ends, promoting its polymerisation into nucleoprotein filaments responsible for the homology search and strand-exchange reactions[19–21]. RAD51 delivery to a DSB is mediated by the BRC repeats, an array of eight related BRCA2 motifs that can bind RAD51 independently and with varying affinities[22–24]. The BRC repeats bind RAD51 via

molecular mimicry of its self-association mechanism, sequestering it in a monomeric form suitable for transport[25,26].

Our mechanistic understanding of BRCA2's replicative role lags behind that of its participation in HDR. Current evidence shows that BRCA2 cooperates with RAD51 to preserve the structural integrity of the fork, by restraining the MRE11 nuclease from excessive degradation of newly synthesised DNA[12,27]. Protection of nascent DNA involves RAD51-dependent fork reversal[28] and stabilisation by BRCA2[12] of the RAD51 filament that forms on the regressed arm of the reversed fork. Filament stabilisation is mediated by the specific interaction of the RAD51-binding TR2 motif located in the BRCA2 C-terminus[4,12,29]. Deletion of TR2-encoding *BRCA2* exon 27 reduces lifespan in mice and causes genomic instability in cells exposed to DNA damaging agents[30,31]; furthermore, the TR2 motif helps protect stalled and collapsed replication forks[32]. The importance of TR2 is highlighted by its reinstatement in reversion mutants of *BRCA2*-deficient cancer cells that have become resistant to treatment[33].

Biochemical evidence shows that TR2 recognises the multimeric state of RAD51 and associates with RAD51 nucleoprotein filaments in an interaction that is inhibited by CDK phosphorylation of TR2[34–36]. Thus, TR2 and BRC repeats exert contrasting biochemical effects on RAD51, leading to current models whereby TR2 stabilises RAD51 filaments and protects them from disassembly by BRC repeats[34,35]. However, our

[1]Department of Biochemistry, University of Cambridge, Cambridge CB2 1GA, UK. ✉e-mail: lp212@cam.ac.uk

understanding of the mechanism by which the BRCA2 TR2 motif binds RAD51 and protects filaments lacks a structural basis. Here we have used cryoEM and structure-guided mutagenesis to elucidate the mechanism of TR2 binding to RAD51 nucleoprotein filaments. Our findings provide a structural rationale for RAD51 filament stabilisation by BRCA2, a critical interaction for the protection of replication forks and maintenance of genomic integrity.

## Results

### A conserved acidic patch in RAD51 mediates BRCA2 TR2 and BRC4 binding

An exposed acidic patch in fission yeast Rad51, located on a short alpha helix adjacent to the protomer interface of the filament, was shown recently to act as an interaction hub for recombination mediators Rad52, Rad55-Rad57 and Rad54[37] (Fig. 1A). To investigate whether the same site in human RAD51 mediates the interaction with the BRCA2 TR2 motif, we generated a RAD51 protein carrying a single D184A or double D184A, D187A mutation, that reduced or neutralised the acidic nature of the patch (Fig. 1B and Supplementary Fig. 1).

We assessed the interaction of the RAD51 acidic-patch mutants by electrophoretic mobility shift analysis (EMSA) of RAD51 nucleoprotein filaments (NPFs) in the presence of a BRCA2 TR2 peptide spanning amino acids 3260 to 3308. TR2 binding causes stabilisation of RAD51 NPFs and formation of filament aggregates[34,35]: titration of TR2 on wild-type RAD51 NPFs showed the expected enhancement of filament formation accompanied by formation of larger species of reduced electrophoretic mobility (Fig. 1C, Supplementary Table 1). Filaments formed with the single D184A or double D184A, D187A RAD51 mutants showed a reduction in TR2 binding proportional to the decrease in charge at RAD51's acidic patch (Fig. 1C); the reduction was especially pronounced for filaments formed on ssDNA.

A known biochemical property of TR2 is the protection of RAD51 nucleoprotein filaments from disassembly by BRC repeat 4 of BRCA2[34,35]. To test whether the acidic patch was also required for BRC4 binding to RAD51, we examined whether the interaction of a BRC4 peptide spanning BRCA2 amino acids 1514 to 1548 was impacted by the D184A and D184A, D187 mutations. The ability of BRC4 to disrupt RAD51 NPF formation was moderately reduced for the single D184A mutant and severely impaired for the D184A, D187A double mutant (Fig. 1D, Supplementary Table 1). The reliance of BRC4 binding on RAD51's acidic patch is supported by the crystal structure of the RAD51–BRC4 complex[26], which shows that RAD51 D187 forms a polar network of contacts with BRC4 residues T1526, S1528 and K1530 in the beta hairpin that mimics RAD51 FxxA self-association motif (Supplementary Fig. 2).

We investigated further the effect of the acidic patch mutations on TR2 binding using surface plasmon resonance (SPR). We developed an SPR assay whereby increasing amounts of TR2 peptide were added to RAD51 NPFs that had been assembled on biotinylated DNA coupled to a streptavidin chip (Supplementary Fig. 3A, Supplementary Table 1). Both RAD51 mutants displayed a clear reduction in TR2 binding, with the double D184A, D187A mutant showing weaker binding compared with the single D184A mutant (Fig. 1E, Supplementary Fig. 3B). As observed by EMSA, the impact of the mutations was stronger for NPFs formed on ssDNA (Fig. 1E). We used the same SPR assay to assess the effect of the mutations on the ability of BRC4 to disrupt an established filament (Supplementary Fig. 4A, Supplementary Table 1). Whereas BRC4 was able to induce disassembly of a wild-type RAD51 NPF, this ability was moderately reduced with the single D184A mutant and to a large degree with the double D184A, D187A mutant (Fig. 1F, Supplementary Fig. 4B).

This evidence shows that the acidic patch of human RAD51 mediates binding of both RAD51-binding motifs in BRCA2, the BRC repeats and TR2, as observed for binding of recombination mediators to fission yeast Rad51[37].

### BRCA2 TR2 induces bundling of RAD51 filaments

To understand the structural basis for the association of the BRCA2 C-terminus with the RAD51 nucleoprotein filament, we sought to determine the structure of a RAD51-DNA-TR2 filament by high-resolution cryo-electron microscopy (cryoEM). Incubation with the BRCA2 TR2 peptide led to extensive bundling of RAD51 NPFs with both ss- and dsDNA (Fig. 2A, Supplementary Table 1). Individual filaments within a bundle appeared to be in translational register relative to each other as it was possible to obtain well-resolved 2D classes of filament pairs (Supplementary fig. 5A, Supplementary Table 1). Filament aggregation induced by TR2 complicated picking of individual filament particles and prevented structural determination by cryoEM. We therefore sought to develop a method to prepare RAD51 NPFs in the presence of TR2 peptide without causing filament bundling. We reasoned that bundling might be alleviated by capping the DNA substrate with a bulky tag, to prevent filament aggregation by steric interference. We therefore prepared ss- and dsDNA substrates with a biotin tag at either one end or both ends, that could be bound by monomeric streptavidin (mSA) (Supplementary Fig. 5B, Supplementary Table 1). RAD51 NPFs reconstituted on DNA capped at both ends with mSA were competent to bind the TR2 peptide (Fig. 2B). Importantly, RAD51 NPF reconstitution with mSA-capped DNA yielded filaments that showed no bundling in the presence of TR2 (Fig. 2C, Supplementary Table 1).

### CryoEM structures of RAD51 NPFs bound to BRCA2 TR2

Using biotinylated DNA capped with mono-streptavidin we were able to prepare RAD51 NPF samples in the presence of a stoichiometric excess of TR2 peptide, to maximise occupancy of its filament binding sites while avoiding bundling. We obtained RAD51 NPF structures on both ss- and dsDNA with bound TR2 (Supplementary Figs. 6–8; Supplementary Table 2), which showed clear density attributable to the TR2 peptide (Fig. 3A). TR2 decorates the outside surface of the filament (Fig. 3B), binding across adjacent RAD51 protomers (Fig. 3C). BRCA2 residues 3289 to 3304 could be modelled in the map at the TR2-RAD51 interface, with no appreciable difference in binding mode between filaments on ss- and dsDNA (Supplementary Fig. 9).

BRCA2 TR2 adopts a bipartite mode of binding to the RAD51 filament. TR2 residues 3296-KAFQPPR-3302 bind in extended conformation to an exposed groove on the surface of RAD51 ATPase domain that includes the acidic-patch helix (Fig. 3C). The polar side chains of K3296, Q3299 and R3302 project towards the acidic patch, forming a basic stripe that complements the charge of RAD51 D184 and D187 (Fig. 3D). Hydrophobic contacts are provided by residues F3298 and P3301 that become buried at the interface (Fig. 3E); the aromatic side chain of F3298 packs against the peptide bond linking RAD51 E154 and G155.

RAD51's acidic-patch helix leads into the ATPase domain's outermost beta strand, which pairs up with the inter-domain linker of the proximal RAD51 protomer in the filament. TR2 amino acids 3289 to 3295 extend beyond the acidic-patch groove and reach the protomer interface, to interact with the adjacent RAD51 molecule in the filament (Fig. 3C). Residues 3292-PAAQ-3295 fold in a single-turn alpha helix that packs against RAD51's inter-domain linker sequence, shielding the RAD51 residues that mediate self-association between protomers in the filament (Fig. 3F). S3291, which is a target of CDK phosphorylation[36], is in the correct position to hydrogen bond to the glutamate of RAD51 E91 (Fig. 3F).

### Structure-based mutational analysis of the RAD51–BRCA2 TR2 interaction

The local resolution at the TR2 site was lower than for the global map, which complicated the model fitting for the peptide. Fitting with the correct sequence register was guided by the density of the large side chains in the highly conserved TR2 region 3296-KAFQPPR-3302

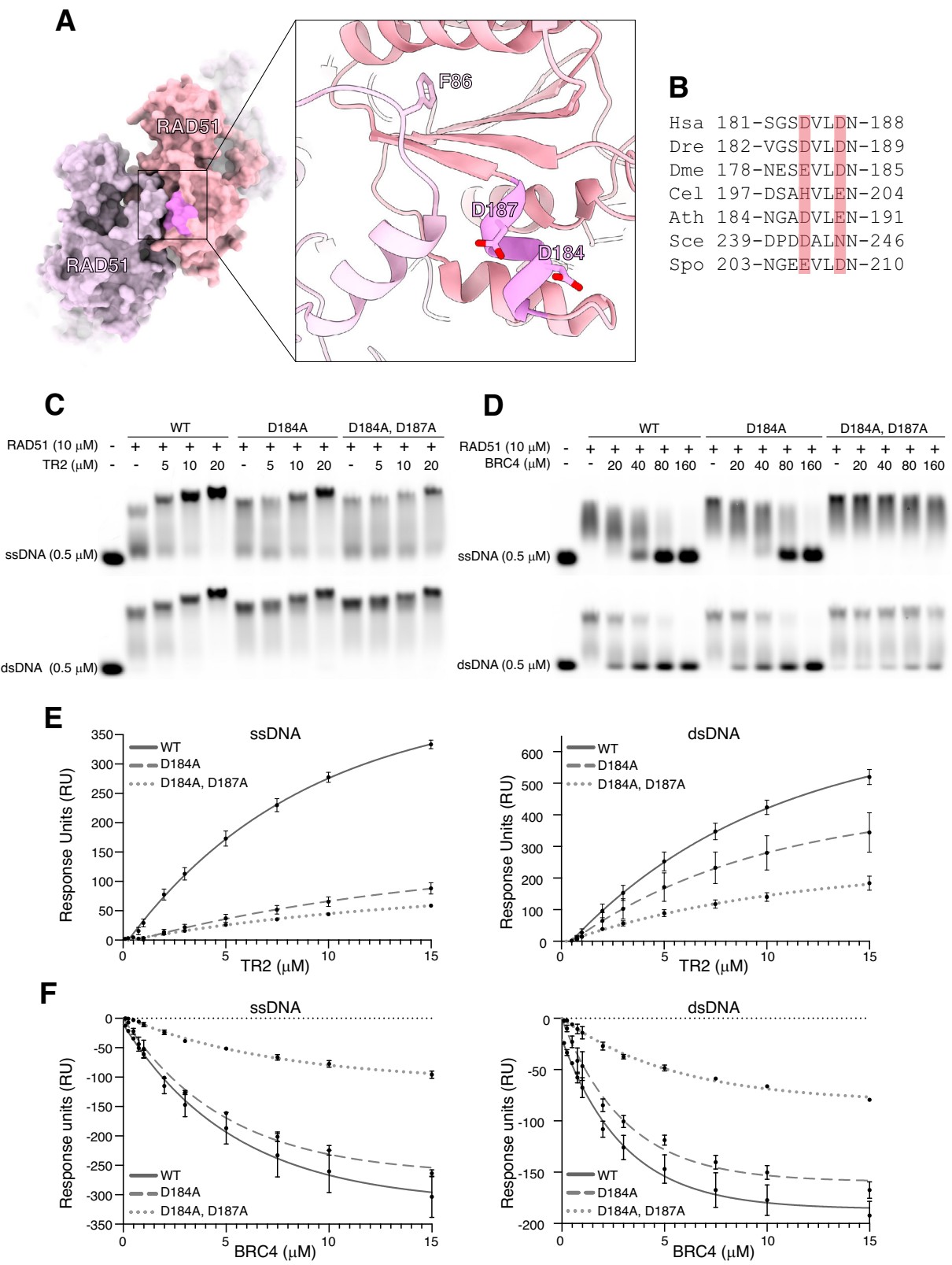

(Fig. 3A). To validate our molecular model of BRCA2 TR2 association with the RAD51 NPF, we performed a mutational analysis of the RAD51–TR2 interface, targeting residues that are predicted by the structure to be important for the RAD51-TR2 interaction (Fig. 4A, B). We generated TR2 mutants as Maltose-Binding Protein (MBP) fusion proteins (Supplementary Fig. 1) and measured by EMSA their ability to bind the RAD51 NPF.

To assess the importance of TR2's polar interactions, we generated alanine mutants of basic-patch residues K3296A, Q3299A and R3302A that reduced its positive charge, as well as triple alanine (3A) and aspartate (3D) mutants that targeted all three positions, to abolish or reverse their polar charge respectively. Single-point TR2 mutants K3296A and R3302A showed a modest decrease in filament binding whereas the Q3299A mutant exhibited increased association with the

**Fig. 1 | An acidic patch on the RAD51 surface mediates binding of BRCA2 TR2 and BRC4. A** Molecular surface representation of the RAD51 NPF (PDB ID: 8BQ2) with two adjacent RAD51 molecules coloured thistle and light pink, and the acidic patch coloured in brighter pink. The inset shows the the position of the alpha helix bearing acidic residues D184 and D187, as well as F86 of the adjacent RAD51 molecule, which is important for RAD51 self-association in a filament. **B** Multiple sequence alignment of acidic patch residues in RAD51 (Hse: *Homo sapiens*; Dre: *Danio rerio*; Dme: *Drosophila melanogaster*; Cel: *Caenorhabditis elegans*; Ath: *Arabidopsis thaliana*; Sce: *Saccharomyces cerevisiae*; Spo: *Schizosaccharomyces pombe*). The position of D184 and D187 in the alignment is highlighted. **C** Electrophoretic mobility shift assay of TR2 peptide titrations on RAD51 NPFs reconstituted with wild-type, single D184A or double D184A, D187A mutant protein and ss- or dsDNA. DNA was visualised by 473 nm excitation of the fluorescein label. ssDNA experiment performed twice, dsDNA experiment performed once. **D** Same assay as in **C**, but with the BRC4 peptide. BRC4 was incubated with RAD51 prior to addition of ss- or dsDNA. Both experiments performed once. **E** Steady-state binding curves of surface plasmon resonance (SPR) measured upon TR2 peptide injection on immobilised RAD51 NPFs coupled to a streptavidin chip via biotin-tagged DNA. NPFs comprised wild-type, single D184A or double D184A, D187A RAD51 and ss- or dsDNA. Data are presented as mean values ± SEM, $n = 3$ independent experiments. **F** Same SPR assay as in **E** but titrating the BRC4 peptide on RAD51 NPFs. Data are presented as mean values ± SEM, $n = 3$ independent experiments. Source data are provided as a Source Data file.

filament (Fig. 4C, Supplementary Table 1). The 3A mutant disrupted the TR2 interaction with the filament almost entirely and the 3D mutant abolished binding completely (Fig. 4D, Supplementary Table 1).

S3291 phosphorylation or mutation to alanine or glutamate had been shown to block the interaction of TR2 with RAD51[36]. The structure shows that S3291 contributes a polar contact to the glutamate side chain of RAD51 E91 (Fig. 3E). Although TR2 S3291A binding to the RAD51 NPF was drastically reduced, the alanine mutant was still able to associate with the filament (Fig. 4E, Supplementary Table 1). The TR2 motif engages in hydrophobic interactions with RAD51 that are mediated by F3298 and P3301. To probe the importance of these hydrophobic contacts for the association of TR2 with RAD51, we assessed the impact of F3298A on the interaction. Alanine mutation of TR2 F3298 abolished TR2 binding to the RAD51 NPF (Fig. 4E).

The structure of RAD51 NPF with bound TR2 shows that RAD51 S181 comes in close proximity to TR2 C3304 (Fig. 5A). To further validate our model of the RAD51–TR2 interaction, we mutated RAD51 S181 to a cysteine to enable disulphide formation with TR2 C3304, which would support its presence at the TR2-RAD51 interface, as identified in the structure. Under mild oxidation conditions we could readily detect formation of a crosslinked RAD51–TR2 species in the sample of the RAD51 S181C mutant in the presence of TR2 (Fig. 5B, C).

## Discussion

Here we have described the structural basis for binding of the BRCA2 C-terminus to the RAD51 nucleoprotein filament, a critical interaction for maintenance of genomic integrity by the BRCA2 tumour suppressor protein. Our results provide a rationale for the biochemical observation that BRCA2 TR2 stabilises the RAD51 NPF, as TR2 binding across the protomer interface acts as a molecular brace for the association of juxtaposed protomers in the filament.

Our evidence shows that TR2 and BRC4 rely on RAD51's acidic patch residue D187 for binding (Supplementary Fig. 10). Whereas RAD51 D187 participates in a network of hydrogen bonds with BRC4 hairpin residues T1526, S1528 and K1530[26], its role in the interaction with TR2 is to provide charge neutralisation of the polar groups in K3296, Q3299 and R3302 that likely assists the correct local TR2 folding. The observation of an overlap in the binding footprint of TR2 and BRC4 provides a structural basis for the known ability of the TR2 motif to protect RAD51 NPFs from disassembly by BRC4[34,35]. Remarkably, the same acidic patch mediates the interaction of recombination mediators with fission yeast Rad51[37], pointing to an evolutionarily conserved mechanism of recruitment.

Our data provide a molecular basis for the inhibition of RAD51 binding caused by CDK phosphorylation of BRCA2 S3291[36]. S3291 phosphorylation disrupts its interaction with RAD51 E91 and likely causes local TR2 unfolding, destabilising its association with RAD51. The TR2 S3291A mutant retains a degree of binding to the filament (Fig. 4E), which bears on its use as a separation-of-function mutant for comparative studies of BRCA2 function in recombinational repair and replication[12,38,39]. Our data shows that a better mutant to use for such studies would be F3298A, which causes a complete loss of TR2 binding (Fig. 4E).

Human BRCA2 binds the meiotic recombinase DMC1—a closely related RAD51 orthologue—via the PhePP motif in the conserved sequence 2404-KVFVPPFK-2411[40]. The PhePP motif is highly similar to the TR2 motif 3296-KAFQPP-3302 that we identified here as critical for RAD51 binding. This observation indicates that BRCA2 adopts a common mechanism mediated by a KxFxPP motif to interact with RAD51 and DMC1.

Our cryoEM maps let us model a highly conserved portion of BRCA2 TR2 that spans residues F3289 to C3304. TR2 sequence conservation extends N-terminally to our model, up to K3263. Recent evidence showed that TR2 harbours a DNA-binding activity centred on residues 3266-KKRR-3269[41], indicating that TR2 contain spatially distinct DNA- and RAD51-binding activities in its N- and C-terminal halves, respectively. We were unable to detect additional density for TR2 in the vicinity of either ss- or dsDNA, indicating that RAD51-bound TR2 does not engage with DNA within the same filament. Indeed, the biochemical propensity of TR2 to induce filament bundling likely results from simultaneous engagement in trans with RAD51 and DNA. We note that the TR2 sequence linking its DNA- and RAD51-binding motifs is rich in prolines and therefore likely to adopt an extended conformation, as expected of a physical spacer. Such TR2 properties might underpin BRCA2's ability to promote pre-synaptic RAD51 filament formation, as well as filament pairing with target dsDNA in HDR and at stalled forks (Fig. 6).

RAD51 is an established cancer drug target[42]. Our structure pinpoints the groove on the RAD51 surface that accommodates TR2 residues 3296-KAFQPPR-3302 as potentially suitable for the development of small-molecule compounds that disrupt the BRCA2 TR2-RAD51 interaction. Such compounds could be useful in cancer therapy when combined with treatment that induces replicative stress.

## Methods
### Cloning and mutagenesis
Constructs for expression of RAD51 D184A and D184A, D187A mutants were generated by non-overlapping PCR site-directed mutagenesis, using the pMBP4-RAD51 expression vector[43] as a template. The sequences of the mutagenesis primers are reported in Supplementary Table 1.

The construct for expression of the RAD51 S181C mutant was prepared by synthesis (IDT) of a nucleotide sequence corresponding to the G163-to-Stop region of the RAD51 open reading frame and bearing the S181C mutation. The sequence was inserted into the pMBP4-RAD51 vector by restriction digestion with KpnI and AvrII, replacing the wild-type sequence.

An *E.coli* codon-optimised gene fragment encoding residues D3260 to Y3308 of human BRCA2 was synthesised (IDT) with flanking BamHI and HindIII restriction sites and ligated into the pMAT11 expression vector[44] (Addgene plasmid #112592) to express BRCA2 TR2 as a dual-tag His$_6$-MBP (Maltose-Binding Protein) fusion protein. pMAT11 constructs for expression of BRCA2 TR2 mutants S3291A, F3298A, K3296A, Q3299A, R3302A, the triple K3296, Q3299, R3392 alanine (3A) and aspartic acid (3D) mutants were produced in the same way as the wild-type construct.

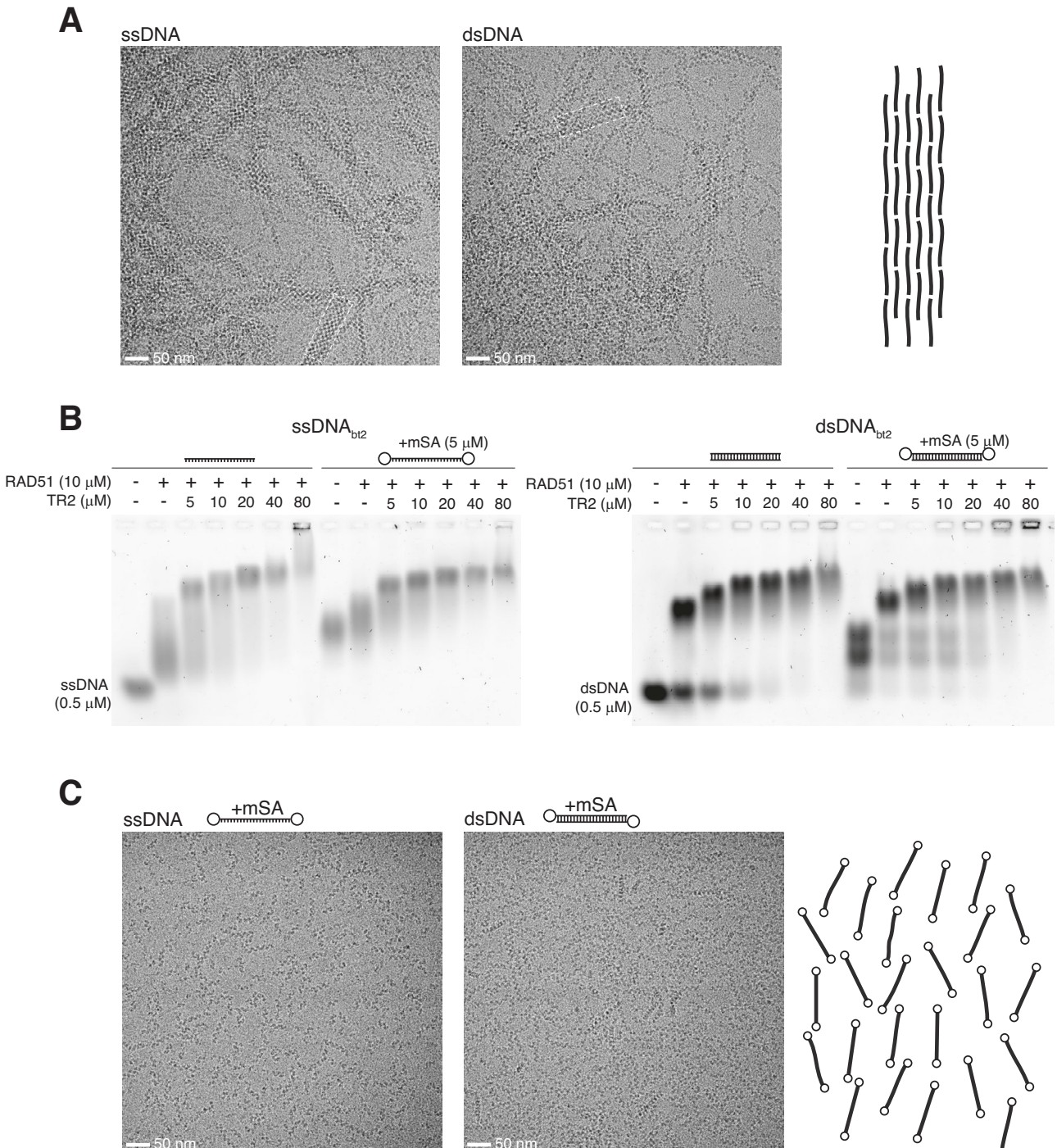

**Fig. 2 | BRCA2 TR2 induces bundling of RAD51 nucleoprotein filaments. A** Cryo-electron micrographs of RAD51 NPFs formed on ssDNA and dsDNA in the presence of 1:0.5 molar ratio of TR2 peptide. Extensive NPF aggregation into bundles of aligned filaments is clearly visible (examples marked with dashed white boxes). Two independent repeats. **B** EMSA analysis of RAD51 binding to doubly-biotinylated ss- and dsDNA, in the absence or presence of mono-streptavidin (mSA). DNA was visualised with SYBR Gold staining. Two independent repeats. **C** Cryo-electron micrographs of RAD51 NPFs in the presence of 1:0.5 molar ratio of TR2 peptide and mSA. Capping of ss- and dsDNA by mSA prevents TR2-dependent filament aggregation. Two independent repeats. Source data are provided as a Source Data file.

## Protein purification

Human wild-type and mutant RAD51 proteins were expressed and purified as previously described[43]. Briefly, human RAD51 was co-expressed with His₆-MBP-BRCA2 BRC4 in *E. coli* BL21(DE3) Rosetta2 strain using the pMBP4-RAD51 vector[43]. Cell pellets were resuspended in 500 mM NaCl, 50 mM HEPES pH 7.4 buffer and, after sonication and centrifugation, the cell lysate supernatant was loaded on a 5-ml HisTrap HP column pre-equilibrated with His Buffer A

(20 mM HEPES pH 7.5, 300 mM NaCl, 5% glycerol) before elution in 40% His Buffer B (20 mM HEPES pH 7.5, 300 mM NaCl, 5% glycerol, 500 mM imidazole). The eluate was diluted in 1 ml batches by mixing with 2 × 1 ml of buffer A125 (20 mM HEPES pH 7.5, 125 mM NaCl, 5% glycerol, 2 mM DTT), loaded on a 5-ml HiTrap Heparin column pre-equilibrated in A125 buffer and eluted with a linear gradient of 12.5% to 100% buffer A1000 (20 mM HEPES pH 7.5, 1 M NaCl, 5% glycerol, 2 mM DTT). Fractions containing RAD51 were

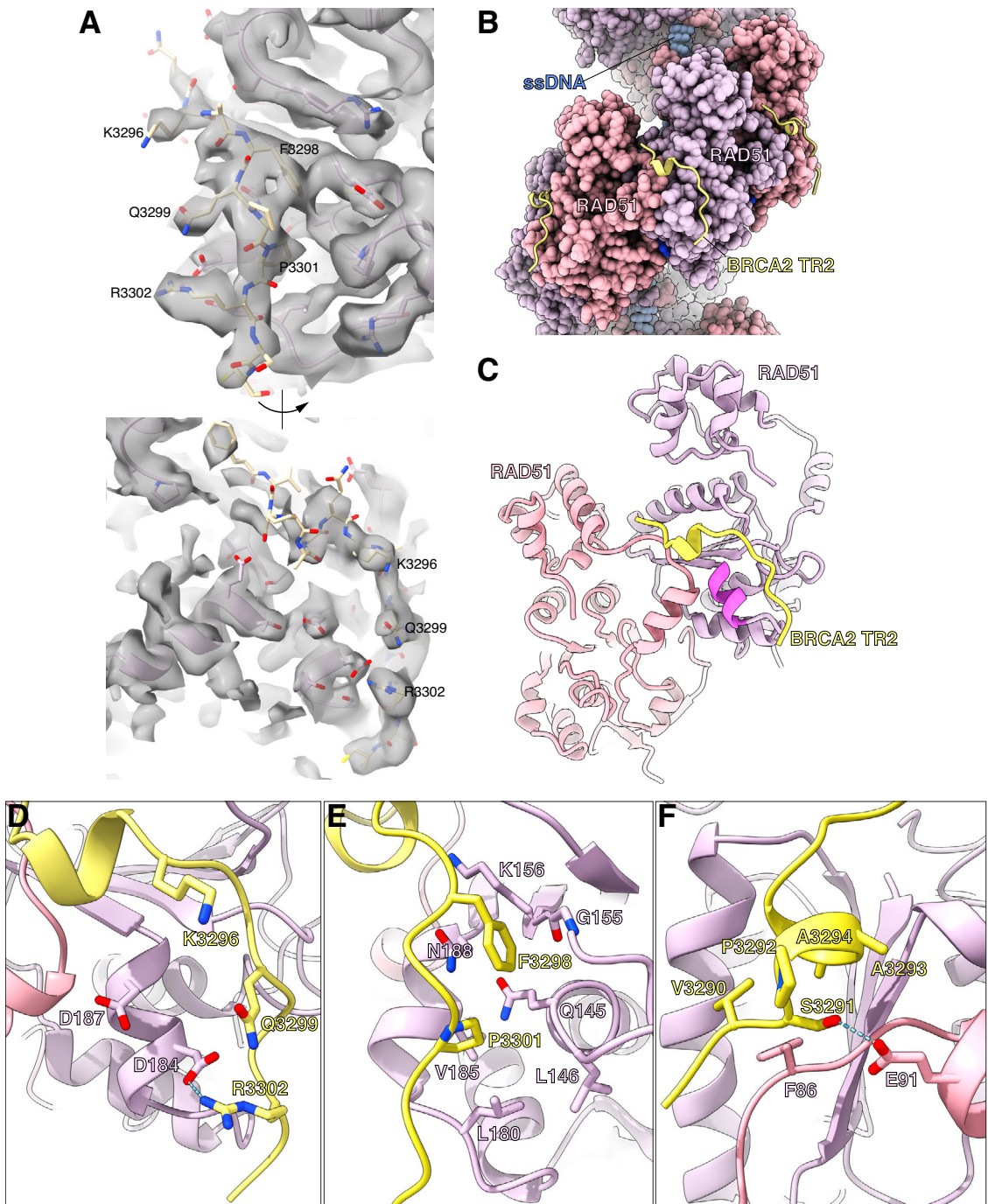

**Fig. 3 | Structural basis for the interaction of BRCA2 TR2 with the RAD51 nucleoprotein filament. A** CryoEM map of the RAD51 nucleoprotein filament on ssDNA with bound BRCA2 TR2. Two rotated views of the map at the TR2 site are shown, with fitted TR2 peptide bound to the RAD51 filament. The side chains of conserved amino acids in the TR2 sequence are labelled for reference. **B** CryoEM structure of the RAD51–ssDNA NPF bound with TR2 peptide. RAD51 and ssDNA are in spacefill representation; RAD51 protomers are coloured alternatively in thistle and light pink. The TR2 peptides that decorate the filament surface are drawn as yellow ribbons. **C** The BRCA2 TR2-RAD51 interface. The TR2 peptide engages the acidic patch of one RAD51 protomer and reaches over to contact the self-association motif of the adjacent RAD51 protomer. RAD51 and TR2 are drawn as ribbons, coloured as in **B**. The acidic patch is shown in pink. **D**–**F** Details of the atomic interactions at the RAD51–TR2 interface. Polar interactions of BRCA2 residues K3296, Q3299 and R3302 with RAD51 D184 and D187 (**D**); hydrophobic contacts of BRCA2 F3298 and P3301 in the groove adjacent to RAD51's acidic patch (**E**); BRCA2's CDK-target residue S3291 becomes partially buried at the TR2-RAD51 interface, where it is hydrogen bonded to RAD51 E91 (**F**).

concentrated and purified further by size-exclusion chromatography using a 16/600 Superdex200 pg (Cytiva) column pre-equilibrated with RAD51 storage buffer (20 mM HEPES pH 7.5, 300 mM NaCl, 5% Glycerol, 2 mM DTT). Purified RAD51 fractions were pooled, concentrated and stored after flash freezing in liquid nitrogen at −80 °C.

All His$_6$-MBP-BRCA2 TR2 fusion proteins were expressed and purified in an identical manner. pMat11-His$_6$-MBP-TR2 constructs were transformed into *E.coli* BL21(DE3)Rosetta2 and transformed cells were used to inoculate 2YT growth medium supplemented with 50 µg/ml ampicillin. Cells were cultured until an OD$_{600}$ reading of 0.7 before induction with 0.5 mM IPTG. Cell pellets were resuspended in

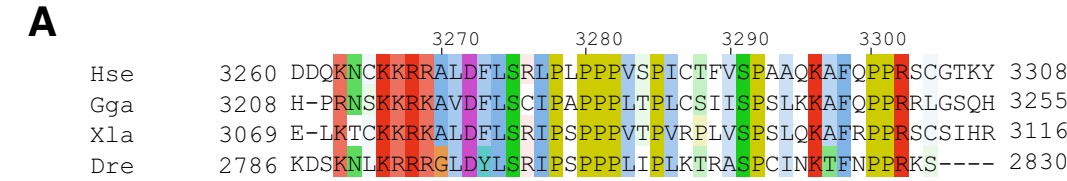

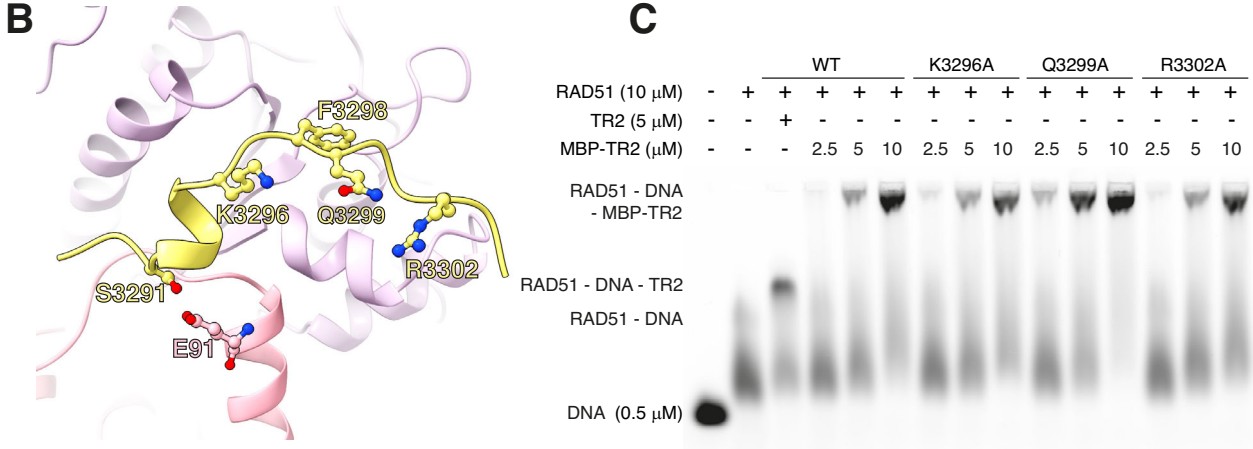

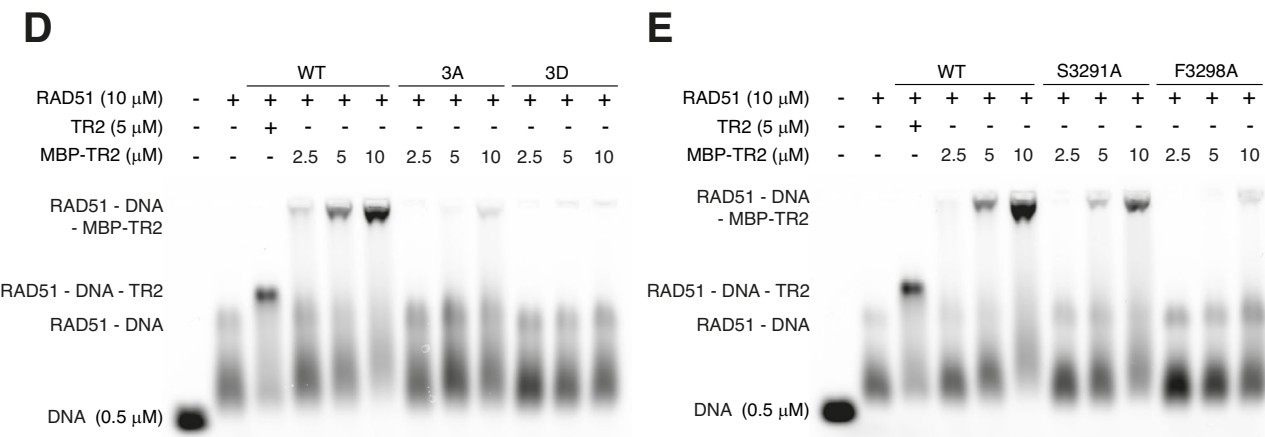

**Fig. 4 | Structure-based mutational analysis of the BRCA2 TR2-RAD51 interface. A** Multiple sequence alignment of BRCA2 TR2 sequences, coloured according to residue type (Hse: *Homo sapiens*; Gga: *Gallus gallus*; Xla: *Xenopus laevis*; Dre: *Drosophila melanogaster*). **B** Drawing of TR2 and RAD51 amino acids targeted for mutagenesis. Drawing style as in D–F of Fig. 3. **C–E** EMSA analysis of RAD51–ssDNA

NPF binding by MBP-fusion TR2 mutants K3296A, Q3299A, R3302A (**C**); 3A (K3296A, Q3299, R3302A), 3D (K3296D, Q3299D, R3302D) (**D**); S3291A, F3298A (**E**). The TR2 peptide was used as a control. EMSA experiments repeated once. Source data are provided as a Source Data file.

resuspension buffer and sonicated to lyse cells in the presence of EDTA-free protease inhibitor. Cell lysate supernatant was loaded on a 5-ml HisTrap HP column pre-equilibrated with His Buffer A before elution in 40% His Buffer B. Fractions containing the His₆-MBP-BRCA2 TR2 protein were pooled and purified further by size-exclusion chromatography using a 16/600 Superdex S75 pg (Cytiva) column pre-equilibrated in storage buffer. To minimise the presence of proteolytically degraded protein, a single fraction at the front of the elution peak containing His₆-MBP-TR2 was retained and concentrated to at least 100 µM before flash freezing in liquid nitrogen and storage at −80 °C.

### DNA oligos
The sequence, size and tag information for the DNA oligos is reported in Supplementary Table 1. All DNA oligos were synthesised and PAGE-purified by IDT and resuspended to 100 µM in TE buffer. For double-stranded (ds) DNA constructs, equimolar concentrations of each

strand were mixed in TE buffer to a final dsDNA concentration of 10 µM and annealed by incubating at 95 °C for five minutes followed by slow cooling to room temperature. Oligos were stored at −20 °C. For the double-stranded biotinylated DNA construct used in SPR, the construct was annealed with a 2-fold excess of the unlabelled strand in TE buffer to a final dsDNA concentration of 10 µM as above.

### Peptides
BRCA2 TR2 and BRC4 peptides were synthesised commercially and dissolved in peptide buffer (100 mM NaCl, 50 mM HEPES pH 7.4) to a concentration of 388 µM (TR2) or 500 µM (BRC4). Peptide sequences are reported in Supplementary Table 1.

### Electrophoretic mobility shift assay
To assess RAD51 nucleoprotein (NPF) formation, wild-type and mutant RAD51 proteins were incubated with DNA in buffer: 25 mM

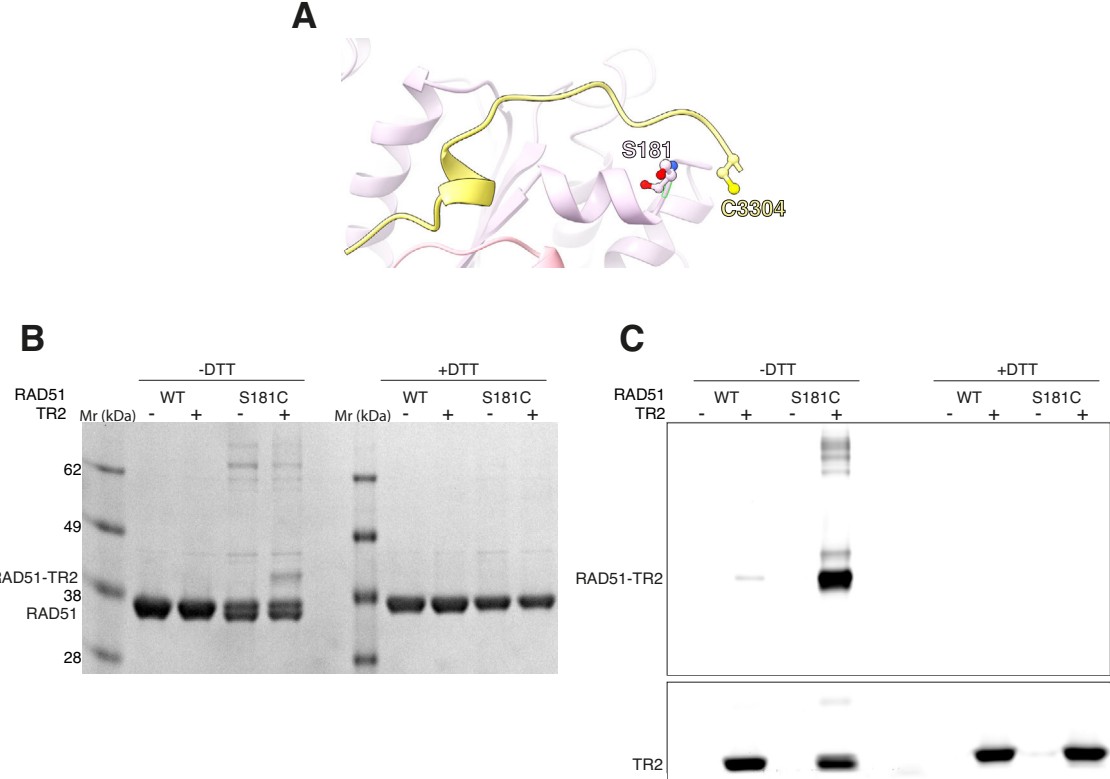

**Fig. 5 | Structure-guided BRCA2 TR2 cross-linking to RAD51. A** Drawing of the RAD51–TR2 interface, highlighting that amino acids BRCA2 C3304 and RAD51 S181 are spatially close, so that the RAD51 S181C mutant can form a disulphide bridge with C3304 in the TR2 peptide. **B** SDS-PAGE analysis of cross-linking reactions between RAD51 S181C and TR2 in the presence of hydrogen peroxide. Wild-type RAD51 and the reducing agent DTT were used as controls. Protein bands were visualised with Coomassie Blue. **C** Same experiment as in **B**, visualised with 532 nm excitation of the Cy3-labelled TR2 peptide. Experiment was repeated three times. Source data are provided as a Source Data file.

HEPES pH 7.5, 150 mM NaCl, 2 mM DTT, 2 mM ATP, 5 mM CaCl$_2$ for 15 minutes at room temperature. For filament binding of the TR2 peptide or His$_6$-MBP-TR2 protein, the RAD51 NPFs were incubated with TR2 for further 5 minutes at room temperature. For filament binding of BRC4, the BRC4 peptide was pre-mixed with RAD51 for 10 minutes at room temperature before addition of DNA. For experiments with mono-streptavidin (mSA)-capped DNA, mSA (Sigma) was added in 5× molar excess of biotin labels for 15 minutes at room temperature prior to the addition of RAD51. Concentrations are provided in figures and DNA sequences for each experiment are reported in Supplementary Table 1.

Before electrophoresis, samples were mixed with a 0.5:1 volume of 50% (w/v) sucrose solution and analysed on a pre-run 0.5% (w/v) agarose gel in 0.5× TB buffer at 35 V for 2.5 hours at 4 °C. DNA was visualised by excitation of the fluorescent tag or, in the case of unlabelled DNA, by SYBR Gold staining. Gels were imaged in a Typhoon FLA9000 by excitation at 473 nm for fluorescein-labelled DNA or SYBR Gold-stained gels, 532 nm excitation for Cy3-labelled DNA or 635 nm excitation for Cy5-labelled DNA.

**Surface plasmon resonance**

All SPR experiments were performed using a Biacore T200 instrument and an S-series SA (Streptavidin) sensor chip. A single chip was used to immobilise biotinylated single- and double-stranded DNA in buffer: 25 mM HEPES pH 7.4, 100 mM NaCl, 1 mM EDTA. ssDNA and dsDNA were immobilised in different channels to a signal of 120 RU and 250 RU, respectively.

All binding experiments were performed in CMA buffer: 25 mM HEPES pH 7.4, 100 mM NaCl, 2 mM DTT, 2 mM ATP, 5 mM CaCl$_2$, 5 mM MgCl$_2$. 1 μM RAD51 (wild-type, D184A or D184A, D187A mutant

proteins) was injected for 120 seconds followed by buffer for 120 seconds at a flow rate of 10 μl/min. TR2 or BRC4 peptides were then injected in a concentration titration series of 0.1, 0.25, 0.5, 0.75, 1, 2, 3, 5, 7.5, 10 and 15 μM for 120 seconds at a flow rate of 10 ml/min followed by a dissociation time of 240 seconds. After each injection of BRCA2 peptide, the sensor chip was regenerated with buffer comprising of CMA buffer with 1 M NaCl for 120 seconds at a flow rate of 100 μl/min. All experiments were performed in triplicate.

For the BRCA2 TR2 titration series, blank-subtracted sensorgrams were used to extract the maximum response accumulated after each TR2 injection. TR2 response accumulation was corrected to account for differences in RAD51 filament formation between runs, by multiplying the maximum response by a scale factor calculated as the ratio between mean RAD51 filament accumulation on each DNA substrate and the mean RAD51 accumulated for each mutant for each triplicate. The first experiment of the triplicate series for binding of TR2 to the D184A, D187A RAD51 NPF on dsDNA was discounted due to a failed peptide injection.

The BRCA2 BRC4 titration series was processed in a similar manner, but response minima were extracted during the peptide injection to account for RAD51 filament disruption. Response minima were corrected in the same manner to account for any differences in RAD51 filament accumulation before peptide injection for each run.

All graph plotting and statistical analysis was performed using Prism 9.3.1 (GraphPad). Response versus peptide concentration graphs show error bars calculated from the triplicates displayed as the standard error of the mean (SEM). Data for each mutant on each substrate were fitted with a non-linear one-phase exponential decay.

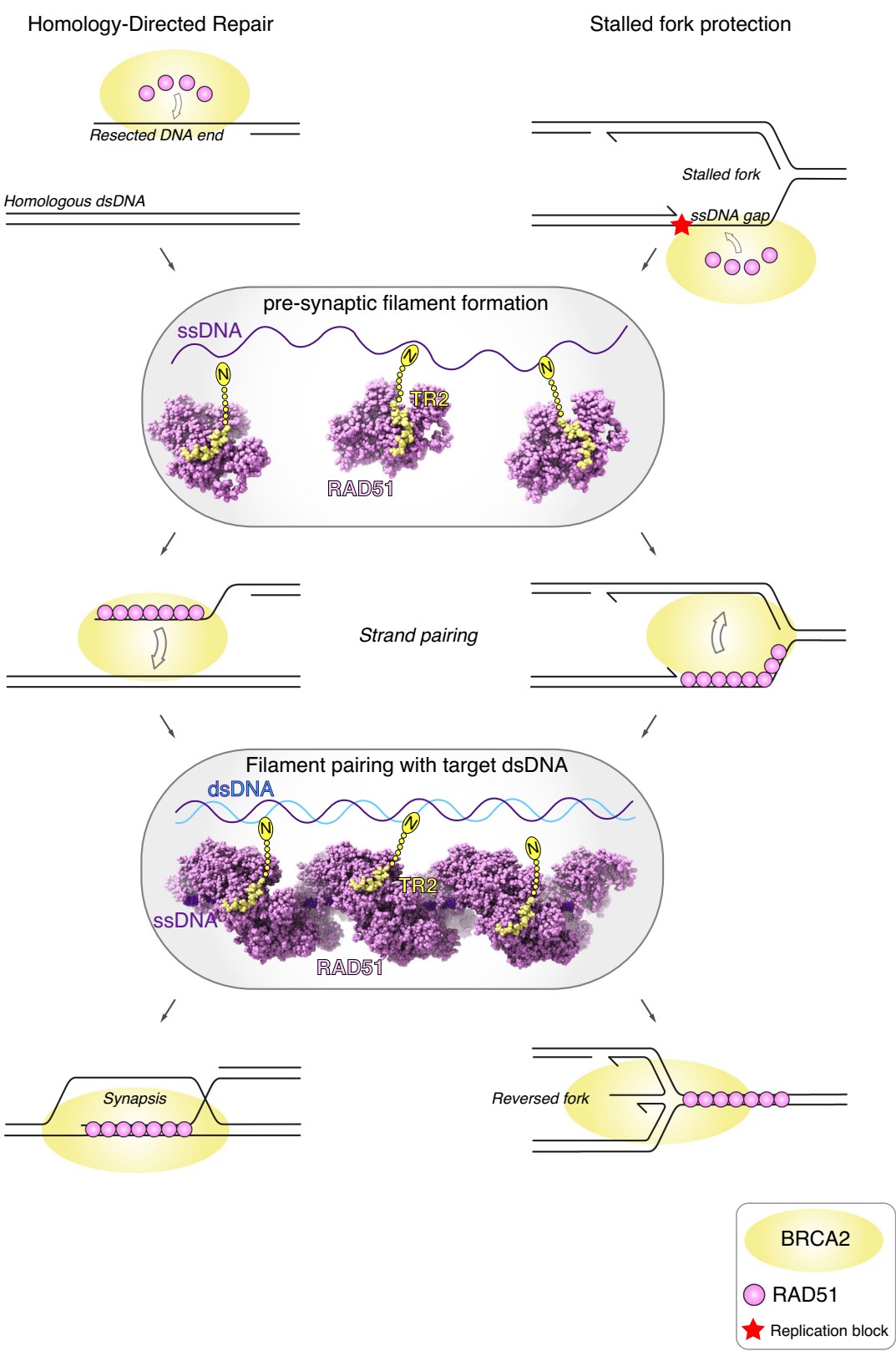

**Fig. 6 | A model for the role of BRCA2 TR2 in homology-directed repair and protection of stalled forks.** The model envisages a possible TR2 intervention at two stages: to promote formation of pre-synaptic filaments on resected DNA ends (HDR) or ssDNA gaps (stalled fork), and to assist in the pairing of the filament with its target homologous dsDNA. In both cases, the underlying mechanism would involve concurrent TR2 engagement in trans with RAD51 and either ss- or dsDNA.

The ovals show spacefill models of RAD51 (pink) with bound TR2 (yellow); the N-terminal sequence of the RAD51-bound TR2, not visible in our structure, has been drawn to illustrate how it could interact with DNA. The model involves the participation of several BRCA2 molecules; the stoichiometry of the BRCA2–RAD51 interaction is currently unknown.

## RAD51–TR2 crosslinking

Cross-linking reactions were prepared in buffer: 25 mM HEPES pH 7.5, 150 mM NaCl, 2 mM ATP, 5 mM CaCl₂, supplemented with 0.5 µM ssDNA. WT or S181C RAD51 was added to a final concentration of 10 µM and incubated for 15 minutes at room temperature. In reactions containing BRCA2 TR2, Cy3-TR2 peptide was added to a final concentration of 1.25 µM and incubated for further 3 minutes. 2 mM hydrogen peroxide was then added and samples incubated for 5 minutes. Each crosslinked sample was analysed by SDS-PAGE, using gel-loading dye with DTT (reducing) or without DTT (non-reducing), on a 12% Bis-Tris SDS-PAGE gel. The gel was first imaged by excitation at 532 nm to visualise Cy3-TR2 on a Typhoon FLA9000 before staining with Coomassie Blue to show total protein content.

## Cryo-electron microscopy (cryoEM)

Mono-streptavidin (mSA)-capped nucleoprotein filaments were reconstituted in buffer: 25 mM HEPES pH 7.5, 150 mM NaCl, 2 mM DTT, 2 mM ATP, 5 mM CaCl₂ supplemented with the appropriate DNA substrate at 250 nM. mSA was first added to a final concentration of 2.5 µM and incubated for 15 minutes at room temperature. RAD51 was then added to a final concentration of 5 µM and the sample was incubated for further 15 minutes to allow for filament formation. The BRCA2 TR2 peptide was added to a final concentration of 40 µM (eight-fold molar excess over RAD51) and incubated for 5 minutes.

UltrAuFoil R1.2/R1.3 300 grids (Quantifoil) were glow-discharged twice for 1 minute using a PELCO easiGlow system (0.38 mBar, 30 mA, negative polarity). 3 µl of sample was applied to each grid before plunge-freezing in liquid ethane using a Vitrobot Mark IV robot (ThermoFisher Scientific), set to 100% humidity, 4 °C, 2 second blot time and −3 blot force.

All cryoEM experiments were carried out at the cryoEM facility of the University of Cambridge in the Department of Biochemistry. CryoEM data collection parameters are reported in Supplementary Table 2. Grids were screened with a 200 keV Talos Arctica microscope fitted with a Falcon III detector (ThermoFisher Scientific) at ×37,000 magnification and with an applied defocus of −4 µM.

**CryoEM of the RAD51-ssDNA-TR2 filament.** A grid of a RAD51 NPF sample reconstituted on doubly-biotinylated mSA-capped single-stranded DNA in the presence of an eight-fold molar excess of BRCA2 TR2 peptide was used to collect cryoEM data on a 300 keV Titan Krios G3 microscope (ThermoFisher Scientific) fitted with a K3 detector (Gatan) using the EPU package. 12,005 micrographs were obtained and processed in Relion3.1[45] at the Cambridge Service for Data-Driven Discovery (CSD3) high-performance computer cluster.

Micrograph motion correction was performed using MotionCor2[46] and CTF estimation was performed using CTFFIND4[47]. Helical autopicking with a picking threshold of 0.1 was performed based on manually-picked 2D class averages and with the number of asymmetric units in the filament box size set to 3 (approximately one half-turn of a filament). Autopicked particles were extracted with a box size of 200 Å and then 2× binned. 2D classification was performed iteratively on extracted particles with the options to ignore CTF correction until the first peak selected and to process using 'fast subsets'.

333,393 high-resolution particles were identified following 2D classification and were used to generate an initial 3D model. 3D classification was performed with the helical reconstruction option turned on and starting values of 56° and 16 Å for helical twist and rise, respectively. All 3D classes showed density for the BRCA2 TR2 peptide and were re-extracted and 2× binned. The 333,393 extracted particles were used to refine class #2 from the 3D classification, which exhibited the clearest TR2 density and the largest proportion of particles, to a resolution of 3.21 Å. The half maps from 3D auto-refine were input to *resolve_cryo_em*[48] from the PHENIX suite[49] to yield a density-modified map at 2.88 Å resolution, which was used for model building.

**CryoEM of the RAD51-dsDNA-TR2 filament.** A grid of a RAD51 NPF sample reconstituted on doubly-biotinylated mSA-capped double-stranded DNA in the presence of an eight-fold molar excess of BRCA2 TR2 peptide was used to collect 10,167 micrographs on the same cryoEM setup as for the RAD51-ssDNA-TR2 NPF.

Data processing was performed in the same way as described for the RAD51-ssDNA-TR2 NPF. 211,532 high-resolution particles were identified by 2D classification, which were used to generate an initial 3D model. 3D classification was performed with the helical reconstruction option turned on and starting values of 56° and 16 Å for helical twist and rise, respectively. All 3D classes except #8 aligned well and were selected for further processing. Selected particles from the 3D classification were re-extracted and binned 2x. The 147,350 extracted particles were used to refine class #1 from the 3D classification, which exhibited the clearest density for the BRCA2 TR2 peptide, to a resolution of 3.62 Å. The half maps from 3D auto-refine were input to *resolve_cryo_em*[48] to yield a density-modified map at 3.34 Å resolution, which was used for model building.

## Model building and refinement

The atomic models of RAD51's pre- (PDB ID: 8BQ2) and post-synaptic (PDB ID: 8BR2) nucleoprotein filaments[50] were docked in the density-modified cryoEM maps of the RAD51-ssDNA-TR2 and RAD51-dsDNA-TR2 filaments, respectively. BRCA2 residues 3289 to 3304 (16 amino acids) could be modelled in the density map of both filaments, with no detectable difference in mode of RAD51 binding. Fitting of the structures in the density-modified maps was optimised with *phenix.real_space_refine*[51] from the PHENIX suite[49], interspersed with minor model rebuilding in Coot[52].

## Reporting summary

Further information on research design is available in the Nature Portfolio Reporting Summary linked to this article.

## Data availability

Coordinates and cryoEM maps for the structure of the RAD51 nucleoprotein filaments with bound TR2 generated in this study have been deposited in the PDB and EMDB, with the accession codes 8PBC, 17584 for ssDNA NPF and 8PBD, 17585 for dsDNA NPF. Requests for materials can be made to the corresponding author. Source data are provided with this paper.

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

## Acknowledgements

We would like to thank Joseph Maman, Tom Blundell and Owen Davies for helpful discussions and advice, Dimitri Chirgadze and staff at the cryoEM Facility of the Department of Biochemistry for help with data collection. This work was funded by Wellcome Trust award 221892/Z/20/Z to L.P. and a BBSRC DTP studentship to R.A.

## Author contributions

R.A. performed all experiments with contributions from K.C. and L.J. L.P. designed and supervised the research. L.P. and R.A. wrote the paper.

## Competing interests

The authors declare no competing interests.
