## [Peer Review File · Nature Communications]

Structural basis for stabilisation of the RAD51 nucleoprotein filament by BRCA2REVIEWER COMMENTS

Reviewer #1 (Remarks to the Author):

The manuscript "Structural basis for stabilization of the RAD51 nucleoprotein filament by BRCA2" by Appleby, Joudeh, Cobbett, and Pellegrini reports the cryo-EM of RAD51 nucleoprotein filament in complex with TR2 peptide (48aa) at the BRCA2 C-terminus. Based on the high-resolution cryo-EM map, they found that TR2 targets an acidic-patch motif near the promoter interface. Based on the structure, they performed mutational analysis on acidic-patch (D184 and D187) of RAD51 and positive residues (K3296, Q3299, R3392) on TR2 using SPR, EMSA and crosslinking to support their structural finding. After reporting this, they suddenly proposed a mechanism of BRCA2's role in HR and stall-fork protection based on the N-terminus of TR2's ability to bind dsDNA and the C-terminus of TR2 binds to RAD51 filament.

Major concern 1

The previous studies, especially a recent Nature Communication publication (2023, 14:432) by Sung group "DNA binding and RAD51 engagement by BRCA2 C-terminus orchestrates DNA repair and replication fork preservation" already showed key interacting residues on TR2 motif using NMR and mutagenesis. The novelty part of this cryo-EM work is to be able to observe specific residue-residue interactions between TR2 motif and RAD51. Although the structure-based mutagenesis work is very solid, both PDB validate reports from their two deposited coordinates indicate that the TR2 peptide has a poor fit to the cryo-EM map. Among these 48 aa TR2 peptide, authors only showed some density of partial peptide (11aa based on PDB validation report with bad fit and low Q score). After examining the density and PDB model, this reviewer believed that, as the PDB validate report indicated, the cryo-EM map of the peptide is not sufficient to reveal the specific residues of TR2 peptide. The density can only reveal that the peptide is near the acidic-patch and a specific interaction between D184 with the peptide. But it is not clear which residue it is solely based on cryo-EM result. Furthermore, There is no interactions observed between D187 and the peptide, however, the D184/D187 double-mutagenesis in this work show differently. Considering that this is helical reconstruction, the cryo-EM map of each protomer and peptide should be similar but, in fact, the maps of individual peptides are different, suggesting those peptides may bind to the acidic-patch in different poses. If so, specific residue-residue interactions shown in this manuscript can be misleading. Overall, the cryo-EM result is not sufficient to identify individual residues of TR2, which defeat the novelty of this work.

Major concern 2

A recent BRAC2 TR2 mechanism was proposed by Sung group (Nature Communications (2023), 14:432) based on very thorough pull-down, DNA binding, HR activities experiments as well as some cellular assays. Another model involved in BRAC2 binding to dsDNA was proposed by Boulton group (Mol. Cell 2023, 83, 1-16) based on single molecule experiment. In Sung's model, BRAC2 is responsible for RAD51 targeting and loading on the DSB site or stalled fork. However, the one proposed in this work takes a further step suggesting BRAC2 not only loads RAD51 but also binds to homologous dsDNA or dsDNA of the stalled fork to facilitate the function of RAD51 NPF. The reason the authors proposed this is based on 1) their finding the TR2 binds to RAD51 NPF, and 2) Sung's findings that KKRR motif (residue 3266-3269) preferably binds to dsDNA. While the BRAC2's function for RAD51 targeting and loading has been widely reported, what is the evidence and need that RAD51 NPF requires BRAC2 to recruit homologous dsDNA during HR process and stalled fork protection? Looking at the model illustrated in Figure 5, authors indicate that BRAC2 binds to several RAD51 promoters within one filament. BRAC2 is a much larger protein compared to RAD51 protomer. It is true that BRCA2 contains multiple motifs (BRC1-8 and TR2) for RAD51 binding. However, only ONE TR2 motif for BRCA2 binding to the RAD51 NPF. In addition, KKRR (residue 3266-3269) is a very small region and may be buried in the BRCA2 protein. What is the evidence that residues 3266-3269 within the content of the whole BRCA2 bind to dsDNA? Even KKRR does, how this ONE dsDNA-binding motif in BRCA2 can help strand pairing of RAD51 NPF? This reviewer would argue that a large protein like BRCA2 remains attached to the RAD51 NPF, it may be likely to interfere the strand pairing by blocking

the entrance of dsDNA. It is not clear (at least to this reviewer) how this proposed model if we consider the stoichiometry and blocking issue.

Major Concern 3:

Lastly, there is a huge missing gap between their TR2-RAD51 binding work and the proposed BRCA2 functional model. To this reviewer, it is not clear what the biological relevance of this model is and what the evidence is to support their model.

Other concerns:

1. Authors should report the local resolution, particularly the TR2 peptide and acidic-patch.
2. For the RAD51 mutagenesis work, authors did not report if the protein quality of RAD51 WT and variants remains the same. Authors should, at least, show the homogeneity of purified RAD51 using SDS-PAGE. In addition, what are the K_m values for WT and variants shown (figure 1e-f and SRP data)? Looking at the curves (figure 1e-1f and extended SRP data), I would argue that the binding constant doesn't change, what the change is the maximal binding capacity to dsDNA/ssDNA upon the presence of TR2 or BRC4. Authors should easily get K_m value from the SPR data shown in the extended data. If K_m doesn't change, this binding data may not support their structural claim.
3. What is the structural explanation for TR2 induce RAD51 filament bundling? For any bundling formation, each TR2 peptide should have at least TWO RAD51 binding motifs or TR2 peptide can self-oligomerize. Does this bundle related to BRCA2 function or just some artifact by using peptide?
4. In result, authors stated "Our results provide a rationale for the biochemical observation that BRCA2 TR2 stabilises the RAD51 NPF as TR2 binding across the promoter interface acts as a molecular brace ...". What is the biochemical observation that BRCA2 TR2 stabilise the RAD51 NPF? If authors refer the observation from figure 1c, can those filaments at high concentration of TR2 be caused by filament bundling shown in their cryo-EM result? Moreover, although cryo-EM map clearly showed that TR2 binds between the protomer interface and TR2 has extensive interactions with one RAD51 promoter, the cryoEM structure ONLY show one hydrogen binding (S3291-E91) between TR2 and the adjacent protomer. How does ONE hydrogen bond to the adjacent promoter provide a structural rationale for NFP stabilisation?

Reviewer #2 (Remarks to the Author):

This is a well-written manuscript that adds to our understanding of BRCA2 through a nice combination of structural biology and biochemistry. In addition, the observation that F3298A completely blocks TR2 binding will be very useful to others.

My comments are all relatively minor.

Science – related

- 1) Line 72: does the BRCA2 – Rad51 complex physically restrain (implying to me at least direct contacts) or simply block MRE11?
- 2) Could the shifts seen on gels reflect bundling as much as formation or stabilization of individual Rad51 filaments? Could this be tested with mutant peptides lacking the KKRR motif proposed to be responsible for bundling?
- 3) Could the hypothesis that the KKRR motif is responsible for bundling be tested in cryoEM using mutant peptides?
- 4) Figure 1: why does there appear to be much less unshifted ssDNA in the Rad51-only lanes of part c vs. part d? Shouldn't they be the same?
- 5) Lines 198 – 204: is the S-S crosslinked species active?
- 6) The methods don't describe cleaving the MBP tag off of Rad51. Was it cleaved off?
- 7) Please move the supplementary figure (s7a) showing the map over the peptide to the main figures.

It was a bit hard to relate the blue density to the peptide structure – it would help to show the map as a transparent surface with the peptide modeled in. Oddly, the blue density appears shorter than the modeled peptide shown in the main figures.

Suggestions to improve clarity

- 1) Specify in the abstract which species of Rad51 this work used
- 2) Line 83: which partner is phosphorylated?
- 3) Explain that BRC4 is one of the BRC repeats of BRCA2 (it may be obvious to those in the field, but it could be taken to mean a different protein).
- 4) "thistle" is a flower not a color. Non-native English speakers might be more familiar with "lavender."

Reviewer #3 (Remarks to the Author):

The review of „Structural basis for stabilisation of the RAD51 nucleoprotein filament by BRCA2“by Appleby et al.

The manuscript includes an exploration of BRCA2 gene's involvement in genomic stability, including DNA double-strand break repair and replication. The authors elucidate increased cancer risk resulting from BRCA2 mutations and introduce its intricate interactions with RAD51 protein., particularly in and replication. The manuscript introduces the BRC repeats' involvement in RAD51 loading as well as the TR2 motif's role in stabilizing RAD51 filaments. The importance of TR2 in protecting replication forks and its connection with RAD51 multimeric states are also highlighted. The results provide insights into the structural mechanism governing the BRCA2 C-terminus interaction with RAD51 nucleoprotein filament. The TR2 motif's role as a stabilizing brace for the filament, and its overlap with BRC4 binding, elucidate TR2's protective function. The authors suggest disruptive effect of CDK phosphorylation of S3291 on the BRCA2-RAD51 interaction, although the exact mechanistic novelty in terms of RAD51 regulation might warrant further clarification. Additionally, the manuscript links TR2's distinct DNA- and RAD51-binding activities with its functional role in DNA repair and replication. Authors effectively bridge structural findings to potential therapies by pinpointing a target groove on RAD51 for drug development. The proposed combination with replicative stress-inducing treatments for cancer therapy enhances the translational value of the study.

However, a few points deserve consideration to enhance the manuscript:

- Incorporating biological data characterising some of the described mutants or more mechanistic characterisation of TR2/BRC competition would enhance the study's depth.
- Explaining the rationale behind RAD51 filament "bundles" formation and its physiological relevance would provide a more comprehensive understanding.
- Addressing the persistence of significant "super" shifts in the RAD51 double mutant (D184A D1871) upon TR titration would clarify the results.
- Direct method monitoring the interaction between RAD51 and TR2 should be included.
- A phospho-mimetic mutant of the TR2 peptide could provide additional support for the regulatory role of S3291.
- In Fig. 5, the model highlights the function of TR2 peptide to bring together the RAD51 nucleoprotein filament with its target homologous DNA. However, this aspect has not been addressed in the manuscript and the work by Kwon et al. (2023) suggest the role of C-terminal binding domain of BRCA2 in timely assembly of the RAD51 filament. This aspect needs further support and clarification. In addition, it should also include the relevance of TR2 and BRC repeat competition.
- Providing the length of DNA used in these experiments would add clarity and reproducibility.

Response to reviewers

We would like to thank the reviewers for their helpful comments and suggestions that have contributed to improve our paper. We have revised the text, figures and supplementary information, and provide additional explanations in the point-by-point responses below.

Reviewer #1 (Remarks to the Author):

The manuscript "Structural basis for stabilization of the RAD51 nucleoprotein filament by BRCA2" by Appleby, Joudeh, Cobbett, and Pellegrini reports the cryo-EM of RAD51 nucleoprotein filament in complex with TR2 peptide (48aa) at the BRCA2 C-terminus. Based on the high-resolution cryo-EM map, they found that TR2 targets an acidic-patch motif near the promoter interface. Based on the structure, they performed mutational analysis on acidic-patch (D184 and D187) of RAD51 and positive residues (K3296, Q3299, R3392) on TR2 using SPR, EMSA and crosslinking to support their structural finding. After reporting this, they suddenly proposed a mechanism of BRCA2's role in HR and stall-fork protection based on the N-terminus of TR2's ability to bind dsDNA and the C-terminus of TR2 binds to RAD51 filament.

Major concern 1

The previous studies, especially a recent Nature Communication publication (2023, 14:432) by Sung group "DNA binding and RAD51 engagement by BRCA2 C-terminus orchestrates DNA repair and replication fork preservation" already showed key interacting residues on TR2 motif using NMR and mutagenesis.

The paper by the Sung lab (Kwon et al, Nat Comm, 2023, published while our paper was in preparation), identifies TR2 F3298 as critical for the interaction with RAD51; our data provides the structural basis for it, by showing that the phenylalanine becomes buried at the TR2 - RAD51 interface. We extend the observation by Kwon et al further, for instance by showing that a triple TR2 mutant of K3296, Q3299, R3302 is impaired for RAD51 binding.

The novelty part of this cryo-EM work is to be able to observe specific residue-residue interactions between TR2 motif and RAD51. Although the structure-based mutagenesis work is very solid, both PDB validate reports from their two deposited coordinates indicate that the TR2 peptide has a poor fit to the cryo-EM map. Among these 48 aa TR2 peptide, authors only showed some density of partial peptide (11aa based on PDB validation report with bad fit and low Q score). After examining the density and PDB model, this reviewer believed that, as the PDB validate report indicated, the cryo-EM map of the peptide is not sufficient to reveal the specific residues of TR2 peptide. The density can only reveal that the peptide is near the acidic-patch and a specific interaction between D184 with the peptide. But it is not clear which residue it is solely based on cryo-EM result.

We respectfully disagree with the reviewer about the uncertainty of the TR2 fit to the map. Our lab has extensive experience in the structural analysis of protein-peptide interactions (see for instance Kilkenny et al, PNAS, 2013; Simon et al, Nature, 2014; Villa et al, Molecular Cell, 2016). We were able to fit unambiguously TR2 residues 3289 to 3304 to the map of the RAD51-TR2 filament on ssDNA, which is at higher resolution relative to the RAD51-TR2-dsDNA filament. Indeed, the reviewer acknowledges that our structure-based mutagenesis work – based on fitting of the TR2 peptide to the cryoEM map – is very solid. The accuracy of the TR2 fitting was further validated by TR2 crosslinking to the S181C RAD51 mutant (Figure 5 of the revised manuscript).

As a general point, it is not unusual for structural analysis of SLIMs (small linear interacting motifs) to show that only a portion of the conserved SLIM sequence becomes ordered upon binding its protein target, see for instance the structure of H2A-H2B bound to FACT (Kemble et al, Molecular Cell, 2016), where only 6 out of 33 conserved amino acids could be visualised in the map of the complex.

Furthermore, There is no interactions observed between D187 and the peptide, however, the D184/D187 double-mutagenesis in this work show differently.

The referee raises the interesting point that, of the two acidic patch RAD51 residues D184 and D187, it is D187 that seems to have a more prominent role in TR2 binding, despite not being in close contact with a specific TR2 residue. We postulate that D187's role in the interaction is predominantly electrostatic, providing the charge neutralisation necessary for TR2 folding in the correct shape for association with the acidic patch of RAD51.

Considering that this is helical reconstruction, the cryo-EM map of each protomer and peptide should be similar but, in fact, the maps of individual peptides are different, suggesting those peptides may bind to the acidic-patch in different poses. If so, specific residue-residue interactions shown in this manuscript can be misleading. Overall, the cryo-EM result is not sufficient to identify individual residues of TR2, which defeat the novelty of this work.

We could not identify meaningful differences in the geometry of the TR2 – RAD51 interaction at different RAD51 protomers. This is as expected, due to the helical averaging employed in the filament reconstruction. Indeed, if

multiple conformations were present, the map would represent an average of the different poses. Importantly, this does not mean that cryoEM maps obtained by helical reconstruction are of the same quality throughout. It is normal for the helical reconstruction to be more ordered towards the centre of the map, due to the way the filament segments are aligned. Indeed, we routinely omit to fit RAD51 protomers at either end of filament reconstructions, as they have worse correlation coefficients with the map.

Major concern 2

A recent BRCA2 TR2 mechanism was proposed by Sung group (Nature Communications (2023), 14:432) based on very thorough pull-down, DNA binding, HR activities experiments as well as some cellular assays. Another model involved in BRCA2 binding to dsDNA was proposed by Boulton group (Mol. Cell 2023, 83, 1-16) based on single molecule experiment. In Sung's model, BRCA2 is responsible for RAD51 targeting and loading on the DSB site or stalled fork. However, the one proposed in this work takes a further step suggesting BRCA2 not only loads RAD51 but also binds to homologous dsDNA or dsDNA of the stalled fork to facilitate the function of RAD51 NPF. The reason the authors proposed this is based on 1) their finding the TR2 binds to RAD51 NPF, and 2) Sung's findings that KKRR motif (residue 3266-3269) preferably binds to dsDNA. While the BRCA2's function for RAD51 targeting and loading has been widely reported, what is the evidence and need that RAD51 NPF requires BRCA2 to recruit homologous dsDNA during HR process and stalled fork protection?

We thank the reviewer for her/his comments, which made us aware of the limitations of our original model of Figure 5. We have now prepared a new version of the figure, which includes a role for TR2 in bringing together RAD51 and ssDNA to facilitate NPF filament formation (Figure 6, top oval in the revised manuscript).

As the reviewer correctly surmises, our model is a direct derivation from the documented biochemical ability of the TR2 motif to interact with both RAD51 and DNA, augmented by our observation that TR2 engagement with RAD51 and DNA likely happens in trans. That BRCA2 might have an additional role, further to promoting NPF formation, in promoting pairing of the RAD51 NPF with its target dsDNA is not a far-fetched concept, as BRCA2 contains multiple domains with demonstrated abilities to bind RAD51 and DNA. Indeed, Kwon et al, Nat Comm, 2023 have shown that TR2 can bind both ss- and dsDNA.

Looking at the model illustrated in Figure 5, authors indicate that BRCA2 binds to several RAD51 protomers within one filament. BRCA2 is a much larger protein compared to RAD51 protomer. It is true that BRCA2 contains multiple motifs (BRC1-8 and TR2) for RAD51 binding. However, only ONE TR2 motif for BRCA2 binding to the RAD51 NPF.

The stoichiometry of the BRCA2:RAD51 interaction in living cells, as well as the number of BRCA2 and RAD51 molecules that co-localise at DNA repair and replication foci, is unknown. A very recent study by the Wyman lab has used live cell imaging to measure a mean number of 25 BRCA2 molecules per DNA repair focus (Paul et al, BioRxiv, 2023). It is therefore possible that multiple BRCA2 molecules might interact with one RAD51 NPF.

In addition, KKRR (residue 3266-3269) is a very small region and may be buried in the BRCA2 protein. What is the evidence that residues 3266-3269 within the context of the whole BRCA2 bind to dsDNA? Even KKRR does, how this ONE dsDNA-binding motif in BRCA2 can help strand pairing of RAD51 NPF? This reviewer would argue that a large protein like BRCA2 remains attached to the RAD51 NPF, it may be likely to interfere the strand pairing by blocking the entrance of dsDNA. It is not clear (at least to this reviewer) how this proposed model if we consider the stoichiometry and blocking issue.

The large size of the BRCA2 protein does not necessarily represent an impediment to the integrated operations of its individual motifs, which would otherwise render the entire protein non-functional. Indeed, BRCA2 should be best considered as a collection of individual domains and motifs residing within the same large polypeptide, that have evolved to promote and mediate RAD51 function. So we don't consider the large size of BRCA2 to be an obstacle to the proposed TR2 function.

Major Concern 3:

Lastly, there is a huge missing gap between their TR2-RAD51 binding work and the proposed BRCA2 functional model. To this reviewer, it is not clear what the biological relevance of this model is and what the evidence is to support their model.

Please see replies above.

Other concerns:

1. Authors should report the local resolution, particularly the TR2 peptide and acidic-patch.

In accordance with the suggestion of the reviewer, we have now added a supplementary figure item (Supplementary figure 8A of the revised manuscript) showing the local resolution of the filament.

2. For the RAD51 mutagenesis work, authors did not report if the protein quality of RAD51 WT and variants remains the same. Authors should, at least, show the homogeneity of purified RAD51 using SDS-PAGE. In addition, what are the K_m values for WT and variants shown (figure 1e-f and SRP data)? Looking at the curves (figure 1e-1f and extended SRP data), I would argue that the binding constant doesn't change, what the change is the maximal binding capacity to dsDNA/ssDNA upon the presence of TR2 or BRC4. Authors should easily get K_m value from the SPR data shown in the extended data. If K_m doesn't change, this binding data may not support their structural claim.

In accordance with the suggestion of the reviewer, we have now added a supplementary figure item showing the SDS-PAGE of the purified proteins used in our study (Supplementary figure 1 of the revised manuscript).

We agree with the reviewer that K_d values should be reported when possible. However, we were unable to achieve binding saturation, necessary for reliable K_d estimation, due to the relatively low affinity of TR2 for RAD51 and consequently the need to inject the TR2 peptide at high concentrations (>30 μ M), which would change sample viscosity and affect the refractive index for detection. The K_d values that we obtained based on the measured range of TR2 concentrations showed the expected trend in affinity ($K_{d_{WT}RAD51} < K_{d_{mut}RAD51}$) but had large confidence interval values that made them statistically unreliable. We therefore decided to report the maximum accumulated response units at each peptide injection, which represent the total amount of peptide binding occurring over the course of injection and is a valid measure of binding affinity. Together with the gel-based evidence of the EMSAs, the SPR data support our identification of the RAD51 acidic patch as a critical mediator of the interaction with BRCA2 TR2 and BRC4.

3. What is the structural explanation for TR2 induce RAD51 filament bundling? For any bundling formation, each TR2 peptide should have at least TWO RAD51 binding motifs or TR2 peptide can self-oligomerize. Does this bundle related to BRCA2 function or just some artifact by using peptide?

We have not investigated in detail the nature of the TR2-dependent filament bundling. The observation that it is disrupted by steric capping of the DNA and that it is possible to derive 2D classes showing a degree of alignment between adjacent filaments suggests that bundling depends on specific interactions that bring filaments together in an orderly fashion. We speculate that it might be caused by concurrent TR2 interactions with RAD51 and DNA of distinct filaments. We believe that it is unlikely that bundling has a direct physiological relevance, but it is an interesting biochemical property that might provide clues as to the mechanism of action of the TR2 motif in regulating RAD51.

4. In result, authors stated "Our results provide a rationale for the biochemical observation that BRCA2 TR2 stabilises the RAD51 NPF as TR2 binding across the protomer interface acts as a molecular brace ...". What is the biochemical observation that BRCA2 TR2 stabilise the RAD51 NPF? If authors refer the observation from figure 1c, can those filaments at high concentration of TR2 be caused by filament bundling shown in their cryo-EM result? Moreover, although cryo-EM map clearly showed that TR2 binds between the protomer interface and TR2 has extensive interactions with one RAD51 protomer, the cryoEM structure ONLY show one hydrogen binding (S3291-E91) between TR2 and the adjacent protomer. How does ONE hydrogen bond to the adjacent protomer provide a structural rationale for NFP stabilisation?

The known stabilising effect of the BRCA2 TR2 on the RAD51 NPF has been described previously, see for instance Esashi et al, "Stabilization of RAD51 nucleoprotein filaments by the C-terminal region of BRCA2." *Nat Struct Mol Biol* **14**, 468–474 (2007). Filament stabilisation is also demonstrated by TR2 binding to wild-type RAD51 NPFs in Figure 1c; please note that the TR2-bound filaments do not show bundling as they are fully able to migrate in the gel.

We chose to highlight the hydrogen bond between TR2 S3291 and E91 RAD51 as S3291 is a known CDK target (Esashi et al, *Nature*, 2005), and our structure provides a structural basis for the role of S3291 phosphorylation. However, this is not the only interaction with the adjacent RAD51 protomer, as TR2 residues 3289 to 3294 also engage in predominantly hydrophobic interactions with the interdomain linker sequence of the adjacent RAD51 protomer, responsible for protomer-protomer association in the filament.

Reviewer #2 (Remarks to the Author):

This is a well-written manuscript that adds to our understanding of BRCA2 through a nice combination of structural biology and biochemistry. In addition, the observation that F3298A completely blocks TR2 binding will be very useful to others.

My comments are all relatively minor.

Science – related

1) Line 72: does the BRCA2 – Rad51 complex physically restrain (implying to me at least direct contacts) or simply block MRE11?

To the best of our understanding, the RAD51 NPF that form at stalled forks prevents excessive DNA degradation by steric exclusion of MRE11; however, this has not been demonstrated explicitly and a specific interaction of BRCA2 and/or RAD51 with the nuclease cannot be ruled out.

2) Could the shifts seen on gels reflect bundling as much as formation or stabilization of individual Rad51 filaments? Could this be tested with mutant peptides lacking the KKRR motif proposed to be responsible for bundling?

3) Could the hypothesis that the KKRR motif is responsible for bundling be tested in cryoEM using mutant peptides?

As mentioned earlier, we don't believe that the bandshift observed for TR2-bound filaments is due to bundling as the filaments are able to migrate into the gels, which would unlikely to be the case given the large size of the filament bundles.

Evidence published by the Sung lab (Figure 2c in Kwon et al, Nat Comm, 2023) shows that the alanine mutation of the KKRR motif abolishes DNA binding. MBP-TR2 lacking the KKRR fails to supershift a RAD51 NPF in the EMSA (unpublished data). We haven't tested the effect of the KKRR motif on filament bundling but, based on this evidence, we would expect it to be required for bundling.

4) Figure 1: why does there appear to be much less unshifted ssDNA in the Rad51-only lanes of part c vs. part d? Shouldn't they be the same?

The RAD51 NPFs formed on short ssDNA oligos are relatively unstable in the biochemical conditions of our EMSAs and small experimental variations can occasionally cause differences in the relative amounts of filament and free DNA. However, this didn't affect the result of the TR2 and BRC4 titrations, which were entirely reproducible.

5) Lines 198 – 204: is the S-S crosslinked species active?

We didn't check whether the cross-linked TR2-RAD51 species was active, as this was beyond the scope of the experiment, which was to validate the observed mode of TR2-RAD51 interaction. We note that crosslinking was promoted by treatment with hydrogen peroxide, which might have caused a degree of oxidation of the protein substrate.

6) The methods don't describe cleaving the MBP tag off of Rad51. Was it cleaved off?

Tag-free RAD51 was co-expressed with 6xHis-HisMBP-BRC4 for purification purposes; the tagged BRC4 protein was used to rescue RAD51 in the first purification step over Ni²⁺-resin and then removed by Heparin chromatography (See Methods).

7) Please move the supplementary figure (s7a) showing the map over the peptide to the main figures. It was a bit hard to relate the blue density to the peptide structure – it would help to show the map as a transparent surface with the peptide modeled in. Oddly, the blue density appears shorter than the modeled peptide shown in the main figures.

In accordance with reviewer's suggestion, we have redrawn the figure, showing the map as a transparent surface with superimposed TR2 peptide (Supplementary figure 8B of the revised manuscript). We would prefer to keep the figure item in the supplementary information to avoid excessive crowding of figure 3.

Suggestions to improve clarity

1) Specify in the abstract which species of Rad51 this work used

We have modified the abstract to clarify that our findings refer to human RAD51.

2) Line 83: which partner is phosphorylated?

We have clarified the text to make clear that TR2 is the target of CDK phosphorylation.

3) Explain that BRC4 is one of the BRC repeats of BRCA2 (it may be obvious to those in the field, but it could be taken to mean a different protein).

We have modified the text accordingly (line 109).

4) "thistle" is a flower not a color. Non-native English speakers might be more familiar with "lavender."

“Thistle” is the name used in ChimeraX. We would rather keep it because it identifies uniquely the colour for any interested reader.

Reviewer #3 (Remarks to the Author):

The review of „Structural basis for stabilisation of the RAD51 nucleoprotein filament by BRCA2“by Appleby et al. The manuscript includes an exploration of BRCA2 gene’s involvement in genomic stability, including DNA double-strand break repair repair and replication. The authors elucidate increased cancer risk resulting from BRCA2 mutations and introduce its intricate interactions with RAD51 protein., particularly in and replication. The manuscript introduces the BRC repeats' involvement in RAD51 loading as well as the TR2 motif's role in stabilizing RAD51 filaments. The importance of TR2 in protecting replication forks and its connection with RAD51 multimeric states are also highlighted.

The results provide insights into the structural mechanism governing the BRCA2 C-terminus interaction with RAD51 nucleoprotein filament. The TR2 motif's role as a stabilizing brace for the filament, and its overlap with BRC4 binding, elucidate TR2's protective function. The authors suggest disruptive effect of CDK phosphorylation of S3291 on the BRCA2-RAD51 interaction, although the exact mechanistic novelty in terms of RAD51 regulation might warrant further clarification.

Additionally, the manuscript links TR2's distinct DNA- and RAD51-binding activities with its functional role in DNA repair and replication. Authors effectively bridge structural findings to potential therapies by pinpointing a target groove on RAD51 for drug development. The proposed combination with replicative stress-inducing treatments for cancer therapy enhances the translational value of the study.

However, a few points deserve consideration to enhance the manuscript:

- Incorporating biological data characterising some of the described mutants or more mechanistic characterisation of TR2/BRC competition would enhance the study’s depth.

Our work was focused on the biochemical and structural characterisation of the specific interaction of BRCA2 TR2 with RAD51, and as such we feel that follow-up biological studies are outside the scope of the current manuscript. We note that our data already provides a structural basis for existing biological findings concerning the role of TR2 in DNA replication, such as those of Schlacher et al, Cell, 2011.

A detailed mechanistic characterisation of the interplay between TR2 and BRC repeats is already available (Esashi et al, NSMB, 2007; Davies and Pellegrini, NSMB, 2007). Our findings provide a structural rationale for these biochemical observations, by showing that binding by both TR2 and BRC4 relies on the same acidic patch residues in RAD51.

- Explaining the rationale behind RAD51 filament “bundles” formation and its physiological relevance would provide a more comprehensive understanding.

We have kept the discussion of filament bundling brief, as we haven’t investigated extensively its mechanism and its biological relevance is uncertain. The dual ability of TR2 to interact with both RAD51 and DNA provides a strong possible molecular basis for concurrent interactions of TR2 with filaments in trans, which would be a logical basis for the bundling effect that we see. We have modified the relevant sentence in the Discussion to make this point clearer, and have also changed the model of Figure 5 accordingly.

- Addressing the persistence of significant “super” shifts in the RAD51 double mutant (D184A D1871) upon TR titration would clarify the results.

Filaments reconstituted with the RAD51 D184A, D187A double mutant protein retain a degree of binding with TR2 but are clearly impaired in their interaction, as shown by the EMSAs in Figure 1C and also by SPR (Figure 1E). This is already mentioned in the relevant portion of the text.

- Direct method monitoring the interaction between RAD51 and TR2 should be included.

We assume that the reviewer is asking for evidence of a direct biochemical interaction between RAD51 and TR2. Such evidence is already available in Davies and Pellegrini, NSMB, 2007; Esashi et al, NSMB, 2007 as well as recently in Kwon et al, Nat Comm, 2023.

- A phospho-mimetic mutant of the TR2 peptide could provide additional support for the regulatory role of S3291.

The importance of S3291 and its phosphorylation on TR2 binding to RAD51 has already been extensively investigated biochemically (Esashi et al, Nature, 2005; Davies and Pellegrini, NSMB, 2007), as well as in cells (Schlachter et al, Cell, 2011). Our findings provide a structural rationale for the role of TR2 S3291 in RAD51 binding.

- In Fig. 5, the model highlights the function of TR2 peptide to bring together the RAD51 nucleoprotein filament with its target homologous DNA. However, this aspect has not been addressed in the manuscript and the work by Kwon et al. (2023) suggest the role of C-terminal binding domain of BRCA2 in timely assembly of the RAD51 filament. This aspect needs further support and clarification. In addition, it should also include the relevance of TR2 and BRC repeat competition.

We thank the reviewer for her/his comments, which made us aware that our model of Figure 5 in its original version offered a reductive interpretation of the range of possible roles of TR2 in HDR and fork protection. We have now prepared a new version of Figure 5 that reflects the possible functions of TR2 in a more balanced way. These include a role in promoting pre-synaptic filament formation by promoting nucleation of RAD51 protomers and recruiting them to ssDNA (top oval in Figure 5), as well as a role in pairing the RAD51 NPF with homologous target DNA. We note that – according to Kwon et al, Nat Comm, 2023 – TR2 possesses both ss- and dsDNA binding abilities, which would be explained by our proposed model.

The reviewer further raises the fascinating point of how the biochemical competition for RAD51 binding by BRC repeats and TR2 would play out in the model. We have chosen to leave out this aspect of BRCA2 function from our drawing to focus on TR2 and avoid excessive speculative interpretations. To the best of our knowledge, we believe that the array of BRCA2 BRC repeats would be responsible for transporting RAD51 protomers to the appropriate DNA site, where the RAD51 cargo would be released; TR2 would then exploit its dual ability to bind both RAD51 and DNA to promote filament formation, as illustrated in the figure.

- Providing the length of DNA used in these experiments would add clarity and reproducibility.

The length of the DNA substrates used in our study are reported in Extended Data Table 1.

REVIEWER COMMENTS

Reviewer #1 (Remarks to the Author):

In the revised manuscript, only major change is to have a new role of TR2 in bringing together RAD51 and ssDNA to facilitate NPF filament formation in the new version of the figure (revised figure 6). This review feels that my previous concerns are not fully addressed.

Previous Concern 1 – cryoEM fitting

The authors disagree about the uncertainty of the TR2 fit to the map. Instead of showing evidence how TR2 reliably fits into the cryoEM map, the authors claimed that their lab has experience in the structural analysis of protein-peptide interactions. Although this reviewer already acknowledged that their binding data is solid. The authors also claimed that it is not unusual that only a portion of peptides can be visualized in the map of the complex. Again, this reviewer did not argue about that, either.

What this reviewer doesn't understand is how this residue 3289 to 3304 is built (not another part of TR2) into a cryoEM map with a local resolution worse than 3.5 angstrom? Even one amino acid shift in the map (eg 3288 to 3303) can greatly change the TR2-RAD51 interactions. As authors insisted that they can fit unambiguously TR2 residue 3289 to 3304, what this reviewer doesn't understand is how someone can fit peptide unambiguously to a low-resolution map region and why Q score from PDB reports says differently. If prior knowledge is applied to assist model building, the related details should be mentioned.

Another concern

As the authors admitted "it is D187 that seems to have a more prominent role in TR2 binding, despite not being in close contact with a specific TR2 residue." They postulated that D187's role in the interaction is predominantly electrostatic, providing the charge. This reviewer could not understand the rationale of "electrostatic, providing the charge"? Both D182 and D187 are charge-charge interactions to TR2 peptide. Isn't D182 also electrostatic and in the closed contact, providing the stronger charge-charge interactions compared to D187. In addition, their postulation is not included in the manuscript.

Previous concern 2- model hypothesis (figure 6)

The major issue, at least for this reviewer, is the supporting evidence for their hypothesis derived from cryoEM structure of RAD51-TR2 filament.

As the authors responded

1. BRCA2 might have an additional role, further to promoting NPF formation, in promoting pairing of the RAD51 NPF with its target dsDNA is not a far-fetched concept, as BRCA2 contains multiple domains with demonstrated abilities to bind RAD51 and DNA as BRCA2 contains multiple domains with demonstrated abilities to bind RAD51 and DNA.
2. The large size of the BRCA2 protein does not necessarily represent an impediment to the integrated operations of its individual motifs, which would otherwise render the entire protein non-functional. Indeed, BRCA2 should be best considered as a collection of individual domains and motifs residing within the same large polypeptide, that have evolved to promote and mediate RAD51 function. So we don't consider the large size of BRCA2 to be an obstacle to the proposed TR2 function.
3. The stoichiometry of the BRCA2:RAD51 interaction in living cells, as well as the number of BRCA2 and RAD51 molecules that co-localise at DNA repair and replication foci, is unknown. A very recent study by the Wyman lab has used live cell imaging to measure a mean number of 25 BRCA2 molecules per DNA repair focus (Paul et al, BioRxiv, 2023). It is therefore possible that multiple BRCA2 molecules might interact with one RAD51 NPF.

This reviewer also believes these claims provide an interesting point and might be possible. However, these claims are mostly hypothetical, requiring some experimental support.

Reviewer #2 (Remarks to the Author):

The authors have satisfied my concerns.

However, I would strongly encourage them to include the density map figure (now in the supplement) in the main text. Readers should be encouraged to judge the map for themselves and hiding it away in a supplement doesn't do that.

I would also encourage the authors to add a close-up over the peptide region of the figure showing the map color coded by resolution (that would be fine in the supplement).

Reviewer #3 (Remarks to the Author):

The revised version of the manuscript continues to offer valuable structural insights into the BRCA2-RAD51 nucleoprotein filament interaction. However, my assessment remains unchanged, as the study, while providing a structural rationale for previously observed biological findings, does not introduce significant novel biological information.

Given this, I believe that the manuscript may find a more appropriate home in a specialized journal with a specific focus on structural biology. This avenue would ensure that the research receives the attention it merits, particularly within the context of its technical and structural aspects.

Response to reviewers

Reviewer #1 (Remarks to the Author):

In the revised manuscript, only major change is to have a new role of TR2 in bringing together RAD51 and ssDNA to facilitate NPF filament formation in the new version of the figure (revised figure 6). This review feels that my previous concerns are not fully addressed.

Previous Concern 1 – cryoEM fitting

The authors disagree about the uncertainty of the TR2 fit to the map. Instead of showing evidence how TR2 reliably fits into the cryoEM map, the authors claimed that their lab has experience in the structural analysis of protein-peptide interactions. Although this reviewer already acknowledged that their binding data is solid. The authors also claimed that it is not unusual that only a portion of peptides can be visualized in the map of the complex. Again, this reviewer did not argue about that, either.

What this reviewer doesn't understand is how this residue 3289 to 3304 is built (not another part of TR2) into a cryoEM map with a local resolution worse than 3.5 angstrom? Even one amino acid shift in the map (eg 3288 to 3303) can greatly change the TR2-RAD51 interactions. As authors insisted that they can fit unambiguously TR2 residue 3289 to 3304, what this reviewer doesn't understand is how someone can fit peptide unambiguously to a low-resolution map region and why Q score from PDB reports says differently. If prior knowledge is applied to assist model building, the related details should be mentioned.

To assist the readers in the assessment of the accuracy of our model building, we have now added a new figure item in the main text (Figure 3A), showing two views of the TR2 model fitted to the map, as well as a closeup view of the density at the TR2 site with fitted peptide in the supplementary information (Figure S8B). In model building the TR2 peptide with the correct amino acid register, we were guided by the density for side chains of residues K3296, F3298, Q3299, P3301 and R3302, that are clearly identifiable in the map (labelled in Figure 3A).

Another concern

As the authors admitted "it is D187 that seems to have a more prominent role in TR2 binding, despite not being in close contact with a specific TR2 residue." They postulated that D187's role in the interaction is predominantly electrostatic, providing the charge. This reviewer could not understand the rationale of "electrostatic, providing the charge"? Both D182 and D187 are charge-charge interactions to TR2 peptide. Isn't D182 also electrostatic and in the close contact, providing the stronger charge-charge interactions compared to D187. In addition, their postulation is not included in the manuscript.

As we explained in the original reply to this point, we think that D187 acts by electrostatic neutralisation of the polar groups in K3296, Q3299 and R3302, and that its role consists in assisting local folding of the TR2 sequence. Charge-charge interaction do not require an exact geometry and can act over a relatively long distance, which would be appropriate for the location of D187 in the RAD51-TR2 interaction. We would also like to point out that D184 does play a role in the interaction, as shown by the SPR data in figure 1E, F. We have added a sentence in the Discussion to highlight the proposed role of D187 in the interaction with TR2.

Previous concern 2- model hypothesis (figure 6)

The major issue, at least for this reviewer, is the supporting evidence for their hypothesis derived from cryoEM structure of RAD51-TR2 filament.

As the authors responded

1. BRCA2 might have an additional role, further to promoting NPF formation, in promoting pairing of the RAD51 NPF with its target dsDNA is not a far-fetched concept, as BRCA2 contains multiple domains with demonstrated abilities to bind RAD51 and DNA as BRCA2 contains multiple domains with demonstrated abilities to bind RAD51 and DNA.
2. The large size of the BRCA2 protein does not necessarily represent an impediment to the integrated operations of its individual motifs, which would otherwise render the entire protein non-functional. Indeed, BRCA2 should be best considered as a collection of individual domains and motifs residing within the same large polypeptide, that have evolved to promote and mediate RAD51 function. So we don't consider the large size of BRCA2 to be an obstacle to the proposed TR2 function.
3. The stoichiometry of the BRCA2:RAD51 interaction in living cells, as well as the number of BRCA2 and RAD51 molecules that co-localise at DNA repair and replication foci, is unknown. A very recent study by the Wyman lab has used live cell imaging to measure a mean number of 25 BRCA2 molecules per DNA repair focus (Paul et al, BioRxiv, 2023). It is therefore possible that multiple BRCA2 molecules might interact with one RAD51 NPF.

This reviewer also believes these claims provide an interesting point and might be possible. However, these claims are mostly hypothetical, requiring some experimental support.

We are pleased that the reviewer consider our points interesting. The model of Figure 6 proposes possible ways in which the TR2 might fulfill its function in recombinational processes during repair and replication, based on the structural evidence in our paper as well as published evidence (Kwon et al, Nat Comm, 2023 for example). As with all models, it contains speculative elements that stretch beyond the existing evidence, with the intended purpose to stimulate further thought and experimentation.

Reviewer #2 (Remarks to the Author):

The authors have satisfied my concerns.

However, I would strongly encourage them to include the density map figure (now in the supplement) in the main text. Readers should be encouraged to judge the map for themselves and hiding it away in a supplement doesn't do that.

In accordance with the recommendation of the reviewer, we have produced a new figure item, Figure 3A in the revised manuscript, showing two view of the TR2 model fitted to the map.

I would also encourage the authors to add a close-up over the peptide region of the figure showing the map color coded by resolution (that would be fine in the supplement).

In accordance with the recommendation of the reviewer, we have added a closeup view of the map coloured according to local resolution at the TR2 site with fitted TR2 peptide, as Figure S8B.

Reviewer #3 (Remarks to the Author):

The revised version of the manuscript continues to offer valuable structural insights into the BRCA2-RAD51 nucleoprotein filament interaction. However, my assessment remains unchanged, as the study, while providing a structural rationale for previously observed biological findings, does not introduce significant novel biological information.

Given this, I believe that the manuscript may find a more appropriate home in a specialized journal with a specific focus on structural biology. This avenue would ensure that the research receives the attention it merits, particularly within the context of its technical and structural aspects.

We respectfully disagree with the reviewer concerning the significance of our findings. A large body of experimental evidence dating back 25 years shows that the TR2 motif has a central role in BRCA2 biology. Our structure provides the first structural basis for its interaction with RAD51, thus helping rationalise a large part of this evidence. Furthermore, we report the unexpected findings that the TR2-binding site on RAD51 is conserved among recombination mediators from yeast to humans, and that the two RAD51-interacting motifs in BRCA2, the BRC repeat and TR2, target the same acidic residue on the RAD51 surface. These findings constitute a major advance in the field and will spur further investigations of the regulatory interactions between RAD51 and its mediators.